# Genetic parameters of growth and adaptive traits in aspen (*Populus tremuloides*): Implications for tree breeding in a warming world

Chen Ding[1¤]*, Andreas Hamann[1], Rong-Cai Yang[2], Jean S. Brouard[3]

**1** Department of Renewable Resources, University of Alberta, Edmonton, Alberta, Canada, **2** Department of Agricultural, Food and Nutritional Science, University of Alberta, Edmonton, Alberta, Canada, **3** Isabella Point Forestry Ltd., Salt Spring Island, British Columbia, Canada

¤ Current address: Western Gulf Forest Tree Improvement Program, Texas A&M Forest Service, College Station, Texas, United States of America

* cd2@ualberta.ca

**Data Availability Statement:** Data collected by the authors on the timing of bud break and leaf senescence are available as S1 Dataset in the supporting information section. However, our

## Abstract

Aspen (*Populus tremuloides* Michx) is a widespread commercial forest tree of high economic importance in western Canada and has been subject to tree improvement efforts over the past two decades. Such improvement programs rely on accurate estimates of the genetic gain in growth traits and correlated response in adaptive traits that are important for forest health. Here, we estimated genetic parameters in 10 progeny trials containing >30,000 trees with pedigree structures based on a partial factorial mating design that includes 60 half-sibs, 100 full-sib families and 1,400 clonally replicated genotypes. Estimated narrow-sense and broad-sense heritabilities were low for height and diameter (~0.2), but moderate for the dates of budbreak and leaf senescence (~0.4). Furthermore, estimated genetic correlations between growth and phenology were moderate to strong with tall trees being associated with early budbreak ($r = -0.3$) and late leaf senescence ($r = -0.7$). Survival was not compromised, but was positively associated with early budbreak or late leaf senescence, indicating that utilizing the growing season was more important for survival and growth than avoiding early fall or late spring frosts. These result suggests that populations are adapted to colder climate conditions and lag behind environmental conditions to which they are optimally adapted due to substantial climate warming observed over the last several decades for the study area.

## Introduction

Trembling aspen (*Populus tremuloides* Michx) is an ecologically and commercially important tree species with high genetic diversity and a broad natural range, including the boreal forest of North America, the eastern United States, and the western mountain ranges from Mexico to Alaska [1, 2]. Aspen can regenerate both via sexual and asexual reproduction [2, 3]. Root

adherence to PLOS policies on sharing data and materials is altered with respect to height and diameter measurements used in this study. These data are owned by an industrial tree breeding cooperative and are therefore not publicly available. Interested researchers may negotiate data sharing agreements with the participating companies, which can be facilitated by Jean Brouard (jean.brouard@ipfgenetics.ca) or the Western Boreal Aspen Corporation, 11420 142 Street NW, Suite #16, Edmonton, AB T5M 1V1 Tel. +1 (780) 482-2795.

**Funding:** Funding was provided by an NSERC/ Industry Collaborative Development Grant CRDPJ 349100-06 to AH and an NSERC Discovery Grant RGPIN-330527-13 to AH through the Government of Canada. The contributing industry partners were Alberta-Pacific Forest Industries, Norbord Inc. (previously as Ainsworth Engineered Canada LP), Mercer Peace River Pulp (formerly Daishowa-Marubeni International Ltd.), Western Boreal Aspen Corporation, and Weyerhaeuser Company Ltd. The funders had no role in study design, data collection and analysis, decision to publish, or preparation of the manuscript.

**Competing interests:** I have read the journal's policy and the authors of this manuscript have the following competing interests: AH received a research grant that included matching financial contributions from industry partners to a government research grant for this study. JSB, representing the consultancy Isabella Point Forestry Ltd., received financial compensations from industry partners for his contributions to experimental design and analysis. The industry partners provided support in the form of research grants to AH and consulting contracts to JSB, but did not have any additional role in the study design, data collection and analysis, decision to publish, or preparation of the manuscript. All data collected by the authors on the timing of budbreak and leaf abscission will be made available through an on-line repository upon acceptance. However, our adherence to PLOS policies on sharing data and materials is altered with respect to height and diameter measurements that were not collected by the authors and that were made available to us by an industrial tree breeding cooperative. These data are not owned by the authors and can therefore not be made publicly available."

suckering often produces large single-species stands after fire disturbances in boreal regions [3,4]. Over the last two decades, aspen has become one of the most important commercial forestry species in western Canada due to hardwood demand from pulp and paper mills, and the development of oriented strand board (OSB) production [5]. Aspen and its hybrids have been utilized in short rotation forestry [6–8]. In Alberta, Canada, aspen tree improvement programs have been developed to maximize the yield in short rotation forestry systems [9,10]. Three geographic breeding regions in Alberta were initially delineated to develop locally adapted and improved planting stock [9], of which two have active tree breeding programs [10].

Successful tree selection and breeding programs depend on sufficiently high heritability for traits of commercial interest. Specifically, additive genetic variance components and narrow-sense heritabilities are of interest to predict the genetic gain from normal recurrent selection. In aspen, dominance and epistatic genetic variance components are of interest as well for clonal selection, because the species can readily be clonally propagated to generate reforestation stock [11]. Broad-sense heritability of height and diameter at breast height (DBH) has previously been estimated in clonal trials and ranges from 0.36 to 0.64, where 262 clones and 11,152 ramets were tested [10]. In a similar experiment, the broad-sense heritability was reported ranging from 0.23 to 0.35 for height and diameter, in which 18 clones and 417 ramets were tested [12]. Heterosis and genotype-by-environment interactions have been studied in juvenile aspen [13,14]. Heterosis of interspecific crosses was also reported for growth and wood quality improvement [9,15].

Estimates of narrow-sense heritabilities, relevant for breeding programs, are usually not available because the additive and non-additive genetic effects are confounded in clonal trials. Also lacking for trembling aspen is the estimate of heritability and genetic correlations of adaptive traits that are important to avoid mal-adaptation and minimize the risk of mortality in plantations [16–18]. For example, unseasonal frost events in spring and fall may damage buds and leaves, and eventually jeopardize productivity and survival [19]. In selection and breeding for tree growth, the inadvertent response of other traits related to fitness may occur as a byproduct. Antagonistic pleiotropy, where one gene controls multiple traits, may play a role in trade-offs between traits that show high negative genetic correlations [20]. Such antagonistic pleiotropy may result in unexpected responses to selection when the correlated response in adaptive traits compromises expected gains in productivity. Pleiotropic loci contributing to phenological traits were reported in *Populus trichocarpa* [21].

Here, we investigate whether improved growth characteristics can be accomplished through tree breeding, while controlling for risks of maladaptation. We evaluate ten progeny trials containing more than 30,000 trees with known pedigree structure, including 60 half-sib families, 100 full-sib families and 1,400 clones to estimate the breeding potential and genetic parameters for collections from Alberta. This paper focuses on the genetic variation within populations and within families which is essential for tree improvement. We estimate additive and non-additive genetic variance components for two growth traits, height, and diameter at breast height, and two adaptive traits, the timing of budbreak and leaf senescence. Further, we estimate genetic correlations among these traits to assess potential trade-offs between growth and adaptive traits. We test the hypothesis that selection for growth may have a correlated response that leads to utilizing a longer growing season, and thereby increases the risk of exposure to late spring frosts or early fall frosts.

## Materials and methods

### Study area and plant material

Active tree improvement programs in western Alberta exist for a northern and a southern breeding region with different climate conditions (Fig 1). The tree improvement programs

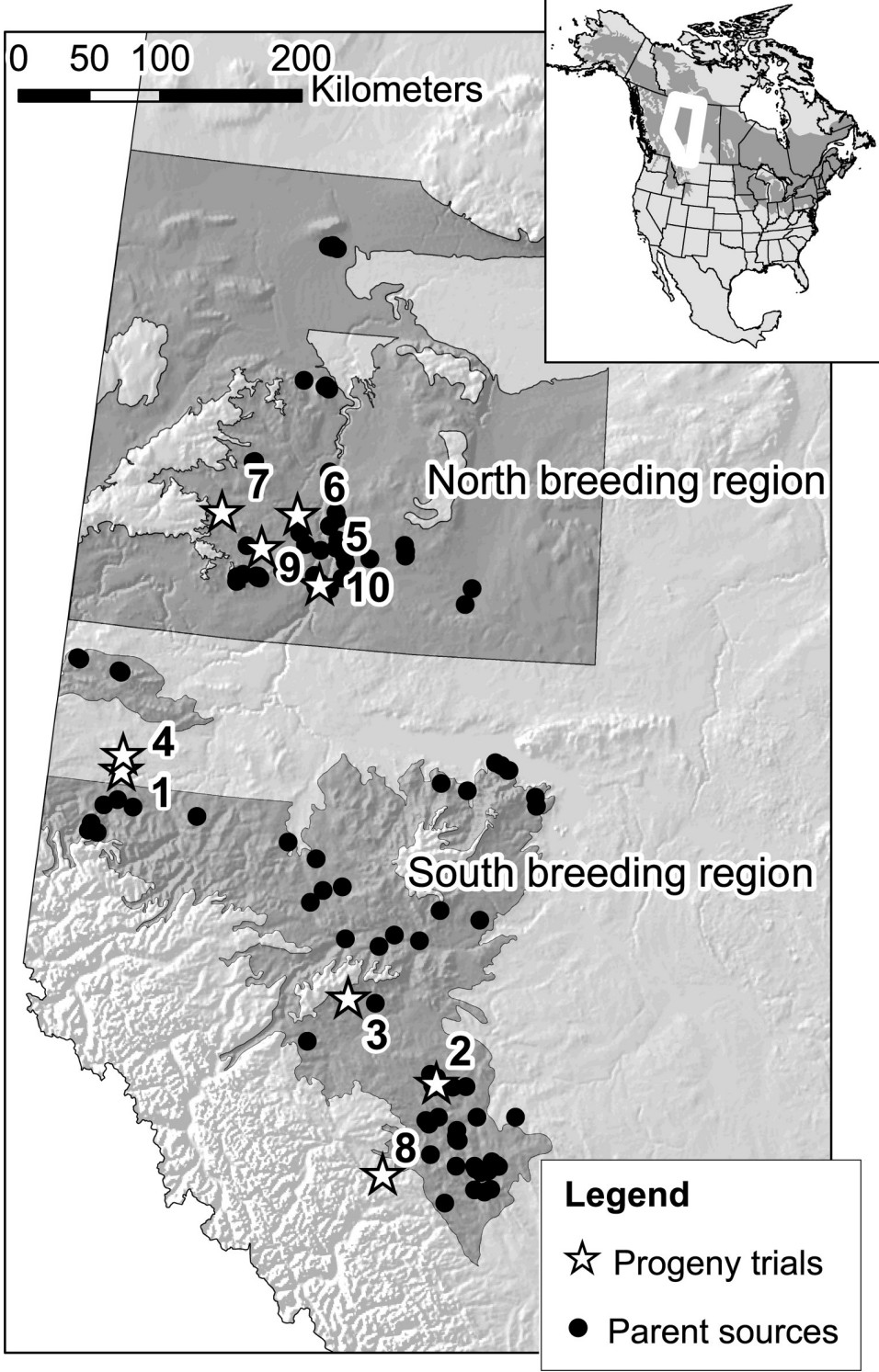

**Fig 1. Study areas in Alberta that were evaluated in this case study.** Circles are the parental sources. Stars and numbers represent the location of trials. Map data was obtained from https://open.alberta.ca/opendata.

initially tested a large number of clones collected from natural stands, targeting plus trees with good form and without signs of pathogens and diseases. In a similar manner, 122 individuals were selected as male or female parents for a partial factorial mating design (to be detailed below). The offspring were planted in ten progeny trials, established between 2005 and 2008 by the Western Boreal Aspen Corporation (WBAC), an industrial collaborative that includes Norbord Inc. (formerly Ainsworth Engineered Canada LP), Mercer Peace River Pulp (previously Daishowa-Marubeni International Ltd.), and Weyerhaeuser Canada Ltd. Detailed origins of the parental material for the progeny trials were previously documented [22].

There were 64 half-sib families as well as 100 full-sib families generated in a partial factorial mating design for the two breeding regions [22]. Each male and female parent was represented by one or two full-sib families and each female parent was also pollinated with a polymix to generate half-sib families. The pedigree structures of northern and southern breeding regions were constructed separately. Trials 1 and 2, planted in 2005, were seedling trials planted in the southern breeding region, sharing the same half-sib and full-sib families. Trials 3 to 8 were planted in 2007 and utilized families from both breeding regions. Families were clonally replicated prior to planting so that these trials have half-sib, full-sib, and clonal structure (i.e., multiple ramets of the same clone, with clones replicating multiple individuals of full-sib families). Trials 9 and 10 were established in 2008, containing different clonal material but overlap in half-sib and full-sib families with trials 3 to 7. These trials were connected through shared half-sib and full-sib families [22].

Seedlings for Trials 1 and 2 were grown in a greenhouse in April to May 2004, hardened in September to October, packed and cold-stored in a refrigerator in winter before planting in May 2005. Clonal planting stock for trials 3–10 was produced over a two-year cycle using a rootling methodology described by Brouard et al. [23]. First, seedlings were grown in 1-gallon pots (3.76 liters) at the Weyerhaeuser Tree Improvement Center in Drayton Valley, Alberta (53˚13'N, 114˚58'W, 869 m) to generate root mass. The following year, the root masses were washed and root cuttings were propagated in Beaver Plastics Styroblock-512 (60 cavities/block 220 ml plug volume) containers at Woodmere Forest Nursery in Fairview, Alberta (56˚04'N, 118˚24' W, 670 m). The resulting rootling ramets were then hardened, packed and cold-stored in a refrigerator in winter before planting in May 2007 and 2008 [23].

## Experimental design and phenotypic measurements

All trials were constructed using an alpha design [24]. The use of alpha designs allowed for the flexible allocation of treatment and block numbers, and advantage over conventional randomized incomplete block designs. The location, exact experimental design, numbers of half-sib and full-sib families and clones for each trial are provided in Table 1. Over 30,000 individual trees were planted in this progeny trial series. Border trees surrounded each trial, and all trials except 7 and 10 were fenced to prevent browsing. Mulch layers or brush blanket mats were used to control competing vegetation, and spacing was 3×3m.

Height measurements were carried out multiple times between 2005 and 2013 with extendable measuring poles, and diameter at breast height (DBH) was assessed with a diameter tape. Budbreak scores were obtained based on a repeated scoring method according to Li et al. [25]. Score 0 was recorded for dormant buds, score 1 indicated a swollen bud, score 2 indicated broken bud scales, score 3 was given for the emergence of green leaves, score 4 indicated leaf extension, score 5 indicated more than two leaves emerged, and score 6 indicated fully unfolded leaves. Scores were recorded for each individual tree on April 12, 20, 22, 24, 26, May 1, 3, 9, 11, 13, 15, 17, 19 in 2010 at trial 2. At trial 3, scores were recorded on April 18, 21, 25,

**Table 1. Locations, experimental design, family structure, clonal structure and measurement averages for sapling height, DBH, survival, day-of-the year (DoY) of budbreak, and day-of-year (DoY) of leaf senescence (leaf sen.) for ten progeny trials of *Populus tremuloides* in Alberta.** Note that Trials 1 & 2 do not have clonal replications of genotypes.

| Trial | Est. year | Latitude | Longitude | Elevation (m) | Experimental Design[1] | Half-sibs | Full-sibs | No of clones | No of trees | Height (m) | DBH (cm) | Surv. (%) | Budbreak (DoY) | Leaf sen. (DoY) |
|---|---|---|---|---|---|---|---|---|---|---|---|---|---|---|
| | | Southern Breeding Region | | | | | | | | | | | | |
| 01 | 2005 | 55˚60' | -120˚48' | 662 | 6×10×9×3 | 33 | 51 | - | 1,620 | 4.77 | 4.80 | 72 | - | - |
| 02 | 2005 | 53˚18' | -116˚30' | 962 | 6×10×9×3 | 33 | 50 | - | 1,620 | 3.27 | 3.30 | 69 | 128 | 267 |
| 03 | 2007 | 53˚48' | -115˚30' | 968 | 9×24×24×1 | 37 | 83 | 560 | 5,184 | 1.36 | 5.50 | 70 | 135 | 268 |
| 04 | 2007 | 55˚12' | -120˚48'' | 808 | 9×20×25×1 | 36 | 73 | 508 | 4,500 | 1.02 | | 64 | - | - |
| 08 | 2007 | 52˚42' | -116˚00' | 1,234 | 9×8×6×1 | 2 | 28 | 47 | 432 | 0.70 | | 71 | 135 | 273 |
| | | Northern Breeding Region | | | | | | | | | | | | |
| 05 | 2007 | 56˚24' | -118˚48' | 525 | 9×20×24×1 | 33 | 71 | 471 | 4,320 | 1.32 | 3.70 | 71 | - | - |
| 06 | 2007 | 56˚48' | -118˚24' | 570 | 9×21×21×1 | 32 | 77 | 455 | 3,969 | 2.00 | | 65 | - | - |
| 07 | 2007 | 56˚48' | -119˚36' | 850 | 9×8×6×1 | 2 | 27 | 47 | 432 | 0.52 | | 18 | - | - |
| 09 | 2008 | 56˚36' | -118˚06' | 650 | 9×21×20×1 | 31 | 61 | 491 | 3,780 | 2.01 | 1.70 | 81 | - | - |
| 10 | 2008 | 56˚24' | -118˚48' | 525 | 9×21×20×1 | 32 | 53 | 459 | 3,780 | 0.40 | | 50 | - | - |

[1]) The experimental design is described as the number of: complete blocks × incomplete alpha blocks within complete blocks × treatments within alpha blocks × trees per treatment in a row plot. The maximum number of treatments (clones or families) in the experiment is determined by the number of alpha bocks × treatments within each alpha block. However, the actual number of tested clones or tested families may be smaller, with filler trees or additional treatment replications filling the gaps. Est. year, year of establishment.

27 and May 2, 10, 12, 14, 16, 18, 20 in 2010. In trial 8, scores were recorded on April 11, 18, 23, 26, May 3, 9, 10, 13, 15, 17, 19, 21 in 2010.

For leaf senescence scores, scoring was based on an eight-level scale according to Fracheboud et al. [26]. Score 0 represented uniformly green leaves, score 1 indicated darker than pale green leaves, score 2 indicated a majority of pale green leaves, score 3 indicated more green than yellow leaves, score 4 indicated a majority of yellow leaves, score 5 indicated only yellow leaves, score 6 indicated 20% brown leaves, score 7 was 50% leaf abscission, and score 8 represented ≥90% leaf abscission. Observation dates for fall phenology in trial 2 were September 1, 7, 21, 29 October 5, 20 in 2011. Scores in trial 3 were recorded on September 1, 6, 23, 30 and October 6, 21 in 2011. Trial 8 was assessed on August 31, 7, 21, 29 September and October 20, 2011.

The day-of-year of a phenology event (DoY) was subsequently calculated for a critical score that showed the best separation among genotypes: score 3 for spring phenology (emergence of green leaves) and score 7 for fall phenology (50% leaf abscission). The date when the phenology reached the critical score of individual trees was determined by either the first record of the critical score, or by linear regression from the bracketing dates. The phenology data is provided as S1 Dataset in the supporting information section.

## Statistical and quantitative genetic analysis

All quantitative genetic analysis was conducted with the ASReml-R package [27]. For the seedling trials (1 and 2) we employed the following mixed linear model:

$$\mathbf{Y} = \mathbf{X}\beta + \mathbf{Z}_1 a + \mathbf{Z}_2 f + \mathbf{Z}_3 r + \mathbf{Z}_4 b + \mathbf{Z}_5 p + e \qquad [1]$$

where $\mathbf{Y}$ is the vector of observations of traits (tree height, day-of-year of budbreak or leaf senescence etc.); $\beta$ is a vector of fixed effects; $a, f, r, b$ and $p$ are vectors of additive genetic

effects, full-sib family (cross) effects, replicate effects, block-within-replicate effects and plot effects, respectively; $e$ is a vector of random residuals; $\mathbf{X}$ is the incidence matrix of the fixed effects relating $\beta$ to observations $\mathbf{Y}$; and $\mathbf{Z}_1$ to $\mathbf{Z}_5$ are the incidence design matrices relating the random effects $a, f, r, b$ and $p$ to observations $Y$. We assume that the observation vector follows a normal distribution with the expected value of $E(\mathbf{Y}) = \mathbf{X}\beta$ and with the covariance matrix of $\mathrm{Var}(\mathbf{Y}) = \mathbf{V}$, i.e., $\mathbf{Y} \sim N(\mathbf{X}\beta, \mathbf{V})$. For our data, the $\beta$ vector has only one element (the overall mean). The vectors of five random effects $a, f, r, b$, and $p$, as well as the vector of random residual $e$ are assumed to follow the normal distributions $N(0, \mathbf{A}\sigma_A^2)$, $N(0, \mathbf{I}_f\sigma_f^2)$, $N(0, \mathbf{I}_r\sigma_r^2)$, $N(0, \mathbf{I}_b\sigma_b^2)$, $N(0, \mathbf{I}_{plot}\sigma_p^2)$, $N(0, \mathbf{I}_e\sigma_e^2)$ respectively. Here, $\sigma_A^2$ is the additive genetic variance, $\mathbf{A}$ is the pedigree kinship matrix for describing the additive genetic relationships among individual trees, $\sigma_f^2$ represents 25% of the dominance genetic variance, $\sigma_r^2, \sigma_b^2, \sigma_{plot}^2$ and $\sigma_e^2$ are the variance components corresponding to the vectors of random effects $r, b, plot$ and residual $e$ respectively, $\mathbf{I}_t$ is the identity matrix of order $t$ ($t = f, r, b, plot, e$). Thus, the total variance matrix can be partitioned into components due to the five vectors of random effects described above as well as the residuals,

$$\mathbf{V} = \sigma_A^2 \mathbf{Z}_1 \mathbf{A} \mathbf{Z}_1' + \sigma_f^2 \mathbf{Z}_2 \mathbf{Z}_2' + \sigma_r^2 \mathbf{Z}_3 \mathbf{Z}_3' + \sigma_b^2 \mathbf{Z}_4 \mathbf{Z}_4' + \sigma_{plot}^2 \mathbf{Z}_5 \mathbf{Z}_5' + \sigma_e^2 \mathbf{I}_e \qquad [2]$$

The best linear unbiased estimation (BLUE) of fixed effect ($\beta$) and best linear unbiased prediction (BLUP) of random effects ($a, f, r, b, p$) are solutions to the following mixed model equations,

$$
\begin{bmatrix} \hat{\beta} \\ \hat{a} \\ \hat{f} \\ \hat{r} \\ \hat{b} \\ \hat{p} \end{bmatrix}
=
\begin{bmatrix}
\mathbf{1}'\mathbf{1} & \mathbf{1}'\mathbf{Z}_1 & \mathbf{1}'\mathbf{Z}_2 & \mathbf{1}'\mathbf{Z}_3 & \mathbf{1}'\mathbf{Z}_4 & \mathbf{1}'\mathbf{Z}_5 \\
\mathbf{Z}_1'\mathbf{1} & \mathbf{Z}_1'\mathbf{Z}_1 + \mathbf{A}^{-1}\frac{\sigma_e^2}{\sigma_A^2} & \mathbf{Z}_1'\mathbf{Z}_2 & \mathbf{Z}_1'\mathbf{Z}_3 & \mathbf{Z}_1'\mathbf{Z}_4 & \mathbf{Z}_1'\mathbf{Z}_5 \\
\mathbf{Z}_2'\mathbf{1} & \mathbf{Z}_2'\mathbf{Z}_1 & \mathbf{Z}_2'\mathbf{Z}_2 + \mathbf{I}_f\frac{\sigma_e^2}{\sigma_f^2} & \mathbf{Z}_2'\mathbf{Z}_3 & \mathbf{Z}_2'\mathbf{Z}_4 & \mathbf{Z}_2'\mathbf{Z}_5 \\
\mathbf{Z}_3'\mathbf{1} & \mathbf{Z}_3'\mathbf{Z}_1 & \mathbf{Z}_3'\mathbf{Z}_2 & \mathbf{Z}_3'\mathbf{Z}_3 + \mathbf{I}_r\frac{\sigma_e^2}{\sigma_r^2} & \mathbf{Z}_3'\mathbf{Z}_4 & \mathbf{Z}_3'\mathbf{Z}_5 \\
\mathbf{Z}_4'\mathbf{1} & \mathbf{Z}_4'\mathbf{Z}_1 & \mathbf{Z}_4'\mathbf{Z}_2 & \mathbf{Z}_4'\mathbf{Z}_3 & \mathbf{Z}_4'\mathbf{Z}_4 + \mathbf{I}_b\frac{\sigma_e^2}{\sigma_f^b} & \mathbf{Z}_4'\mathbf{Z}_5 \\
\mathbf{Z}_5'\mathbf{1} & \mathbf{Z}_5'\mathbf{Z}_1 & \mathbf{Z}_5'\mathbf{Z}_2 & \mathbf{Z}_5'\mathbf{Z}_3 & \mathbf{Z}_5'\mathbf{Z}_4 & \mathbf{Z}_5'\mathbf{Z}_5 + \mathbf{I}_{plot}\frac{\sigma_e^2}{\sigma_{plot}^2}
\end{bmatrix}^{-1}
\begin{bmatrix} \mathbf{1}'\mathbf{Y} \\ \mathbf{Z}_1'\mathbf{Y} \\ \mathbf{Z}_2'\mathbf{Y} \\ \mathbf{Z}_3'\mathbf{Y} \\ \mathbf{Z}_4'\mathbf{Y} \\ \mathbf{Z}_5'\mathbf{Y} \end{bmatrix} \qquad [3]
$$

For the trials with clonal single tree plot trials, we modified model (1) into the following linear mixed model:

$$\mathbf{Y} = \mathbf{X}\beta + \mathbf{Z}_1 a + \mathbf{Z}_2 f + \mathbf{Z}_3 c + \mathbf{Z}_4 b + \mathbf{Z}_5 r + e \qquad [4]$$

where the model remains the same as (1) except the plot effect is removed and the effects of clones within full-sib family ($c$) are added. The $c$ factor accounts for the epistasis and ¾ of the dominance [28,29]. We also assumed that $a, f, r, b, p$, and $e$ followed the normal distributions as above respectively; random effect $c$ followed the normal distribution as $N(0, \mathbf{I}_c\sigma_C^2)$.

Narrow-sense and broad-sense heritabilities were calculated based on following functions:

$$\hat{h}_i^2 = \frac{\hat{\sigma}_A^2}{\hat{\sigma}_P^2} = \frac{\hat{\sigma}_A^2}{\hat{\sigma}_A^2 + \hat{\sigma}_{NA}^2 + \hat{\sigma}_e^2} \qquad [5]$$

where $\hat{\sigma}_A^2$ is the additive genetic variance component; $\hat{\sigma}_P^2$ is the phenotypic variance component represented by the sum of $\hat{\sigma}_A^2$, $\hat{\sigma}_{NA}^2$ and $\hat{\sigma}_e^2$; $\hat{\sigma}_{NA}^2$ is the variance of non-additive genetic

effects; the residual error is $\hat{\sigma}_e^2$ [30]. The broad-sense heritability was estimated as:

$$\hat{H}_i^2 = \frac{\hat{\sigma}_G^2}{\hat{\sigma}_P^2} = \frac{\hat{\sigma}_A^2 + \hat{\sigma}_{NA}^2}{\hat{\sigma}_A^2 + \hat{\sigma}_{NA}^2 + \hat{\sigma}_e^2} \tag{6}$$

The standard errors of the heritability were calculated with the delta method [31].

We estimated the additive genetic correlation in seedling trials, genetic correlation in clonal trials ($r_G$), phenotypic correlation ($r_P$) based on individual trees observations. The linear model of tree growth and leaf phenology for a single site is:

$$y_{ijlmn} = t_n + r_{in} + g_{nim} + e_{nimj} \tag{7}$$

where $y_{nimj}$ is the observation of j-th tree of im-th genotype for the n-th trait; $t_n$ represents the $n$-th trait effect, $r_{in}$ is the $i$-th replicate effect of $n$-th trait, $g_{nim}$ is the additive genetic effect of $m$-th genotype of $n$-th trait in $i$-th replicate in the seedling trial, while in the clonal trails, $g_{nim}$ is the genetic effect. In the seedling trial, the genotypes are seedlings nested in replicates, while in the clonal trial, genotypes are clones evenly assigned in each replicate. In seedling trial 2, the genetic correlation is due to the additive genetic effect. In the clonal trials, the genetic correlation is due to the total genetic effect, though the additive genetic effect is more significant, and $e_{nimj}$ is the experimental error for each trial. The fixed effects of the mixed model are similar as the previous model, although the trait effect is added as a fixed factor. The genetic correlation ($r_{G_{ij}}$) was of the form

$$r_{G_{ij}} = \frac{\hat{\sigma}_{G_{ij}}}{\hat{\sigma}_{G_i}\hat{\sigma}_{G_j}} \tag{8}$$

where $\hat{\sigma}_{G_{ij}}$ is the estimated genotypic covariance between trait $i$ and $j$; $\hat{\sigma}_{G_i}$ is the estimated phenotypic standard deviation of trait $i$. For seedling trials the genetic correlation is the additive genetic effect. The phenotypic correlation was calculated as follows:

$$r_{P_{ij}} = \frac{\hat{\sigma}_{P_{ij}}}{\hat{\sigma}_{P_i}\hat{\sigma}_{P_j}} \tag{9}$$

where $\hat{\sigma}_{P_{ij}}$ is the estimated phenotypic covariance between trait $i$ and $j$; $\hat{\sigma}_{P_i}$ is the estimated phenotypic standard deviation of trait $i$.

Based on BLUPs, the breeding value reliability of half-sib parents and individual clones were calculated as follows:

$$R_i = 1 - \frac{PEV}{\hat{\sigma}_i^2} = 1 - \frac{se_i^2}{\hat{\sigma}_A^2} \tag{10}$$

where $R_i$ is the reliability of the breeding value of the $i$-th parent, where PEV is the prediction error variance that equals to the standard error square of the predicted breeding value [32]; and $\hat{\sigma}_A^2$ is the estimated additive genetic variance component.

The correlations of breeding values among sites were calculated as the Pearson's correlation coefficients of half-sibs (breeding values) and clones (genetic values) between trials with *chart. correlation* of the R package *PerformanceAnalytics*. Bootstrapping of correlation coefficients of survival was carried out with the R *boot* package [33]. The G×E effect is explored with the Type-B genetic correlation for tree height, where the same trait measured in two or more environments over the same genetic composition can be treated as two different genetically

correlated traits. R code for estimating genetic parameters according to the models above are available in the Appendix in [22].

## Results

### Genetic parameters of growth traits

Genetic parameters for height and diameter were estimated at a relatively young age. Trees were between 5 and 8 years old at the time of evaluation with the oldest seedling trials having an average height of 3-4m and most clonal trials reaching average heights of 1-2m (Table 1). Dominance and epistatic variance components for height and diameter were small, less than 10% of the phenotypic variance component, so that most broad-sense heritabilities were only marginally higher than narrow-sense heritabilities (Table 2). The highest heritabilities were estimated for a relatively small seedling trial 1 with values around 0.5. For all other trials, heritabilities for height and diameter were quite low (or unreliable) with narrow-sense heritabilities typically ranging from 0.1 to 0.2, and broad-sense heritabilities typically ranging from 0.2 to 0.3.

Type-B genetic correlations based on shared clones and shared full-sib families could only be calculated for sister trials (1–2, 3–4, 5–6, 7–8, and 9–10). Genetic correlations among sister trials yielded $r_{GB}$ values around 0.7 with a standard error of approximate 0.08 for the first three pairs, indicating a relatively low degree of genotype-by-environment interactions ($G{\times}E$). Pairs 7–8 and 9–10 did not yield reliable estimates. Correlations of parental breeding values could be calculated for a larger number of trial pairs that shared parents through the partial factorial mating design. Parent breeding values between sister trials in trials 1 through 6 were generally well correlated, which can be interpreted as low $G{\times}E$. Trial 8 showed a negative correlation with all other trials, and it should be noted that this trial was planted in a relatively cold environment with the highest elevation (Table 1).

### Genetic parameters for adaptive traits

The phenology traits budbreak and leaf senescence were measured at three trials (Table 1), had moderate broad- and narrow-sense heritabilities (Table 3). Narrow-sense heritabilities for budbreak ranged from 0.4 to 0.5, while heritabilities for leaf senescence were slightly lower

**Table 2. Estimates of narrow-sense and broad-sense heritabilities at ten aspen progeny trials for tree height and diameter at breast height (DBH).** Standard errors of the estimates are given in parentheses.

| Trial | Age of Measurement | Narrow-sense heritability ($\hat{h}^2$) | | Broad-sense heritability ($\hat{H}^2$) | |
|---|---|---|---|---|---|
| | | Height | DBH | Height | DBH |
| Southern Breeding Region | | | | | |
| 01 | 8 | 0.55 (0.16) | 0.54 (0.17) | | |
| 02 | 8 | 0.08 (0.10) | 0.03 (0.09) | | |
| 03 | 5 | 0.21 (0.08) | 0.19 (0.07) | 0.33 (0.03) | 0.25 (0.02) |
| 04 | 5 | 0.11 (0.05) | | 0.14 (0.02) | |
| 08 | 5 | No estimate | | 0.03 (0.03) | |
| Northern Breeding Region | | | | | |
| 05 | 5 | 0.10 (0.03) | 0.06 (0.03) | 0.14 (0.02) | 0.09 (0.02) |
| 06 | 5 | 0.13 (0.04) | | 0.20 (0.02) | |
| 07 | 5 | 0.42 (0.45) | | 0.42 (0.22) | |
| 09 | 3 | 0.11 (0.06) | 0.14 (0.05) | 0.19 (0.02) | 0.17 (0.02) |
| 10 | 3 | 0.07 (0.03) | | (0.02) | |

**Table 3. Estimates of narrow-sense and broad-sense heritabilities at three aspen progeny trials for budbreak and leaf senescence.** Broad-sense heritabilities were not estimated for the seedling Trial 02. Standard errors of the estimates are given in parentheses.

| Trial Code | Age of Measurement | Narrow-sense heritability ($\hat{h}^2$) | | Broad-sense heritability ($\hat{H}^2$) | |
|---|---|---|---|---|---|
| | | Budbreak | Leaf senescence | Budbreak | Leaf senescence |
| Southern Breeding Region | | | | | |
| 02 | | 0.46 (0.15) | 0.33 (0.14) | | |
| 03 | | 0.37 (0.10) | 0.42 (0.10) | 0.46 (0.03) | 0.46 (0.03) |
| 08 | | 0.36 (0.07) | no estimate | 0.36 (0.07) | 0.05 (0.05) |

with values between 0.3 and 0.4. Broad-sense heritabilities were only marginally higher than narrow-sense heratibilities, and could not be estimated for the seedling trial 2 (Table 3). Also here, additive genetic variation was most important and dominance and epistatic variance components ranged from zero to 10% of the phenotypic variation. Moderate to strong genetic correlations were found between growth and phenology (r = -0.3 to -0.2 between height and bud break, and 0.6 to 0.8 between height and leaf senescence) with tall trees being associated with early budbreak and late leaf senescence (Table 4). Survival was not compromised by early budbreak or late leaf senescence. In fact, the reverse appeared to be true with negative correlations between survival and budbreak and positive correlations between survival and leaf senescence, just as for growth traits (Table 4).

## Trait variation across the landscape

The correlations between fall phenology and height and survival were generally stronger than those between spring phenology and height and survival, indicating that the adaptive value of fall phenology is high. Neither fall phenology nor budbreak showed strong spatial patterns of breeding values across both breeding regions (Fig 2). However, correlations between height and leaf senescence are visible in these maps: comparing Figs 2 and 3, high breeding values for height (green dots) are often associated with late leaf senescence (pink and purple), with a Pearson's correlation coefficient of 0.65 (p<0.0001).

## Discussion

Positive associations between height and survival as well as increased growth and survival in trees that break bud early and abscise leaves late suggest that utilization of the growing season may be more important than the avoidance of early fall frosts or late spring frosts at all three test sites where phenology was assessed. Strong additive genetic correlations between growth

**Table 4. Estimates of genetic and phenotypic correlations at three aspen progeny trials for budbreak (BUD), leaf senescence (LS), height (HT) and survival (SURV).** Genetic and phenotypic correlations were estimated using an individual tree model for BUD, LS, and HT for the seedling Trial 02, with half-sib families excluded. For the clonal Trials 03 and 08 we used an individual clone model. Phenotypic correlations of survival with all other traits were based on family means (Trial 02) and clone means (Trials 03 and 08) with standard errors determined through bootstrapping.

| | Trial 02 | | Trial 03 | | Trial 08 | |
|---|---|---|---|---|---|---|
| Correlation | Genetic | Phenotypic | Genetic | Phenotypic | Genetic | Phenotypic |
| HT—BUD | -0.30 (0.21) | -0.23 (0.06) | -0.19 (0.05) | -0.25 (0.02) | no estimate | -0.42 (0.05) |
| HT—LS | 0.83 (0.09) | 0.57 (0.04) | 0.58 (0.04) | 0.37 (0.02) | no estimate | 0.28 (0.06) |
| BUD—LS | -0.08 (0.22) | -0.05 (0.07) | 0.15 (0.07) | 0.00 (0.03) | no estimate | -0.20 (0.06) |
| SURV—BUD | | -0.34 (0.09) | | -0.07 (0.06) | | -0.03 (0.20) |
| SURV—LS | | 0.55 (0.11) | | 0.29 (0.04) | | 0.20 (0.12) |
| SURV—HT | | 0.58 (0.11) | | 0.42 (0.04) | | 0.00 (0.15) |

Bud break | Leaf senescence

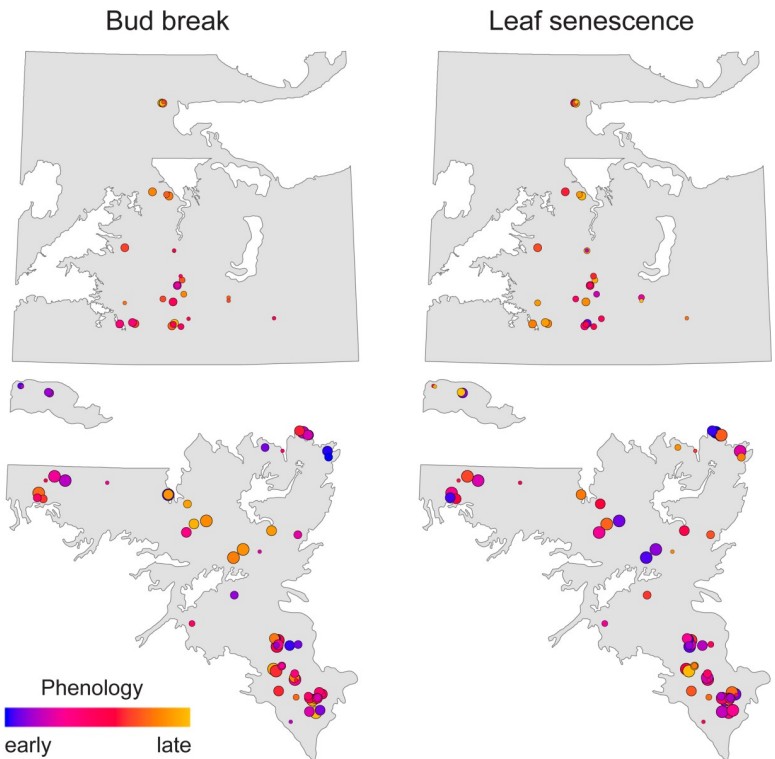

Phenology

early — late

**Fig 2. Breeding values (BV) of parents on southern breeding region test sites for phenology.** The size of the circle indicates the reliability of the estimate (reliability: 0.1 = small circles, 0.7 = large circles). Map data was obtained from https://open.alberta.ca/opendata.

and phenology indicate that much of the genetic gain at the early stage of stand development will be due to expanding the growing season, which may increase the risk of frost damage in spring and fall.

In studies with *Populus* species [34–36] and other species [37], high heritabilities of fall phenology traits, and positive genetic correlations between productivity and fall phenology have previously been observed. Genotypes with a delayed senescence in fall may nevertheless be exposed to increased frost risks. However, climate warming trends that have materialized over the last several decades in Alberta may have decreased the risks of early fall frosts [38]. This might potentially explain our positive association of an extended growing season in fall with high survival. Similar to the expectations of Olson et al. [39], genotypes that utilize a longer growing season may be favored by climate warming at northern latitudes. Inadvertent selection, where individuals with better height and diameter growth are chosen, and the genetically correlated leaf senescence is extended in fall, may therefore be an unplanned but effective climate change adaptation strategy. That said, other climatic factors are likely to change, and while the length of time suitable for growth is likely to continue to increase, water limitations may lead to an overall reduction of boreal forest productivity [40,41]

In contrast, budbreak is a highly plastic trait in response to interannual variation and long-term trends in temperature. Populations can generally be expected to respond appropriately to climate change trends as long as daily temperature variances do not change for given baseline values. In other words, the frost risk associated with a certain heatsum that triggers budbreak needs to remain the same. Yet, the day-of-year for budbreak may have shifted, when this heatsum is reached. Late spring frosts are also considered a more severe threat than early fall frosts,

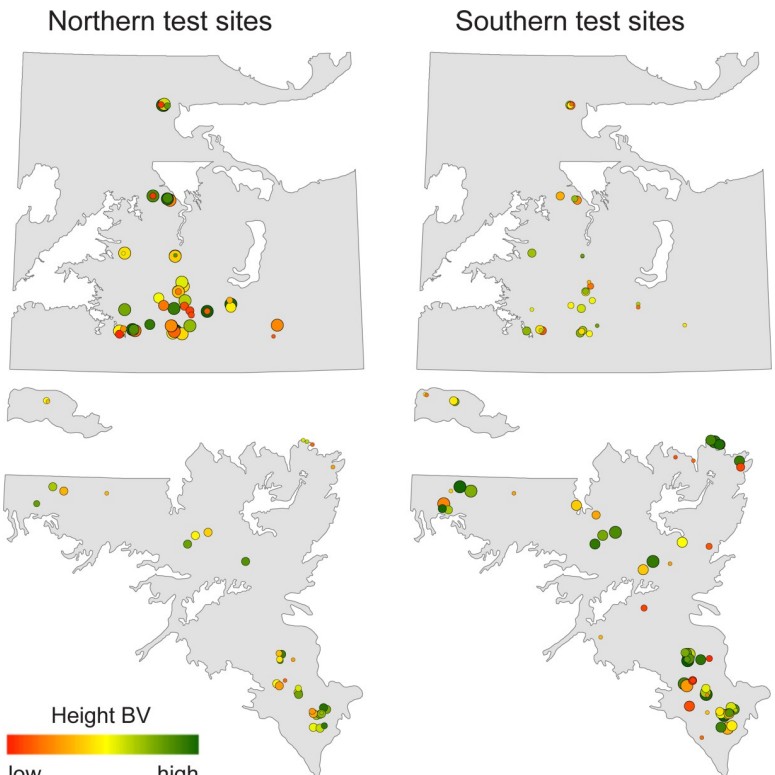

**Fig 3. Height breeding values (BV) of parents tested northern breeding region test sites (left panel) and the southern breeding region sites (right panel).** The size of the circle indicates the reliability of the estimate (reliability: 0.1 = small circles, 0.7 = large circles). The color spectrum of red to green indicating low to high BVs, with values ranging from -55 to 45 cm. Map data was obtained from https://open.alberta.ca/opendata.

as they may destroy buds and juvenile leaves and severely compromise early-season growth [42]. In our study, genetic correlations between growth and budbreak were negative (i.e. early budbreak associated with better growth), but they were not nearly as strong as genetic correlations with leaf senescence. Furthermore, survival was not compromised in genotypes that started the growing season relatively early. We, therefore, conclude that correlated selection for early budbreak poses only a small risk. In fact, the results could be interpreted as indicating an adaptational lag with respect to changing climate. The utilization of a longer growing season appears to increase the growth and survival at least in this sample of three field tests. Our within population genetic analysis corroborates results from provenance research that demonstrated adaptational lag among populations through wide-ranging reciprocal transplant experiments [43].

The previous interspecific hybrid breeding mainly exploits heterosis/specific combining ability for growth and wood quality trait improvement [9,15,44]. In our study, intra-specific crosses and within family selection after field selection provide potential gain for growth and predictable phenology response due to the non-additive genetic effect such as specific combining ability and epistasis effect. Instead of producing hybrids with northern species such as *P. davidiana* and *P. tremula* [9], for regional tree improvement, adjacent breeding zones could exchange parentages, crosses, and clones in places like Alberta without severe frost risk concern in spring and fall even in the northern plantation sites [10].

We noted that heritabilities found in this study were generally much lower than those reported in previous studies by Gylander et al. [10] in comparable trial series that investigated

wild clonal selections, and Gylander et al. [10] reported broad-sense heritabilities of 0.51–0.58 from clonal trials. Kanaga et al. [45] Calculated broad-sense heritabilities for growth traits ranging from 0.30 to 0.50 in a short-term common garden study using 13 aspen clones. Our low heritability estimates for growth traits are likely due to strong environmental microsite variation at post-harvest planting sites and the juvenile age at which trees were evaluated. Heritabilities were particularly low for Trials 7 and 10, and only Trial 7 could not be fenced and showed evidence of browsing, while Trial 8 was located at the more elevated site in the study area. For growth selection at multiple sites, results of >8 growing seasons are adequate for mediocre sites and the productive site. We conclude that selection for growth traits at the current stage promises only small to moderate genetic gains, but higher heritabilities may emerge at a later date of trial evaluation.

Genetic correlations among fall phenology and spring phenology traits with growth traits were nevertheless already high in this study, and could further increase as the influence of microsite variation of the planting site decreases with the age of the trees. Survival was also not compromised, but was positively associated with early budbreak or late leaf senescence, indicating that the growing season length was more important for survival and growth than avoiding early fall or late spring frosts. We found that a substantial portion of the tested genotypes are adapted to a shorter growing season than they have experienced during the testing period. Selecting genotypes for reforestation that utilize a longer growing season may be unproblematic under continued climate warming in northern latitudes, and may lead to overall increases in forest productivity as long as water availability under increased evapotranspiration does not become a dominant limiting factor.

## Supporting information

**S1 Dataset. Phenology data for bud break and leaf senescence.** A data table in comma separated values (CSV) format, containing phenology data for Trials 2, 3, and 8. The first 14 rows contain descriptions of the variables.
(CSV)

## Acknowledgments

We thank Jaroslav Klapste for the statistical modelling suggestions. We also thank Earl (Fred) Raley at the Western Gulf Forest Tree Improvement Program for providing help editing the manuscript. We thank Alberta-Pacific Forest Industries, Norbord Inc. (previously as Ainsworth Engineered Canada LP), Mercer Peace River Pulp (formerly Daishowa-Marubeni International Ltd.), Western Boreal Aspen Corporation, and Weyerhaeuser Company Ltd. for providing height, diameter and survival data that has been analyzed in this study, and for providing logistical support for phenology measurements.

## Author Contributions

**Conceptualization:** Chen Ding, Andreas Hamann, Jean S. Brouard.

**Data curation:** Chen Ding, Jean S. Brouard.

**Formal analysis:** Chen Ding, Rong-Cai Yang.

**Funding acquisition:** Andreas Hamann.

**Methodology:** Chen Ding, Rong-Cai Yang, Jean S. Brouard.

**Project administration:** Andreas Hamann.

**Resources:** Jean S. Brouard.

**Visualization:** Chen Ding, Andreas Hamann.

**Writing – original draft:** Chen Ding.

**Writing – review & editing:** Andreas Hamann, Rong-Cai Yang, Jean S. Brouard.

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
