## [Decision Letter · Decision Letter 0]

16 Dec 2019

PONE-D-19-31818

Genetic parameters of growth and adaptive traits in aspen (Populus tremuloides): implications for tree breeding in a warming world

PLOS ONE

Dear Dr DING,

Thank you for submitting your manuscript to PLOS ONE. After careful consideration, we feel that it has merit but does not fully meet PLOS ONE’s publication criteria as it currently stands. Therefore, we invite you to submit a revised version of the manuscript that addresses the points raised during the review process.

Besides meeting the complementary and constructive comments raised by two reviewers, I suggest a bit of effort is put into clarity of tables and figures. I would appreciate it if both origin and planting sites were detailed especially in table 1. It also puzzles me that there is an experimental set up with replicates, but number of replicates (clones) not indicated ... could that number potentially be 1629/6 - or would it be more correct to write 1620 in that column?   Fig 2 presents two phenology traits and one growth trait for the northern garden? whereas figure 3 only presents growth traits for the southern garden. Would it make sense to present only phenology responses in figure 2 and then the two height responses (for the southern and northern origin, I believe) could perhaps be collected in the same figure 3?.

We would appreciate receiving your revised manuscript by Jan 30 2020 11:59PM. To enhance the reproducibility of your results, we recommend that if applicable you deposit your laboratory protocols in protocols.io, where a protocol can be assigned its own identifier (DOI) such that it can be cited independently in the future. For instructions see: http://journals.plos.org/plosone/s/submission-guidelines#loc-laboratory-protocols

We look forward to receiving your revised manuscript.

Kind regards,

Benedicte Riber Albrectsen

Academic Editor

PLOS ONE

Journal Requirements:

**When submitting your revision, we need you to address these additional requirements:**

**Please ensure that your manuscript meets PLOS ONE's style requirements, including those for file naming. The PLOS ONE style templates can be found at http://www.plosone.org/attachments/PLOSOne_formatting_sample_main_body.pdf and http://www.plosone.org/attachments/PLOSOne_formatting_sample_title_authors_affiliations.pdf**Thank you for stating the following in the Acknowledgments Section of your manuscript:Funding was provided by an NSERC/Industry Collaborative Development Grant351 CRDPJ 349100-06 and an NSERC Discovery Grant RGPIN-330527-13 through the Government352 of Canada. We thank Alberta-Pacific Forest Industries, Norbord Inc. (previously as Ainsworth353 Engineered Canada LP), Mercer Peace River Pulp (formerly Daishowa-Marubeni International354 Ltd.), Western Boreal Aspen Corporation, and Weyerhaeuser Company Ltd. for their financial355 and in-kind support.We note that you have provided funding information that is not currently declared in your Funding Statement. However, funding information should not appear in the Acknowledgments section or other areas of your manuscript. We will only publish funding information present in the Funding Statement section of the online submission form.Please remove any funding-related text from the manuscript and let us know how you would like to update your Funding Statement. Currently, your Funding Statement reads as follows:"The funders had no role in study design, data collection and analysis, decision to publish, or preparation of the manuscript."Additionally, because some of your funding information pertains to commercial funding, we ask you to provide an updated Competing Interests statement, declaring all sources of commercial funding.In your Competing Interests statement, please confirm that your commercial funding does not alter your adherence to PLOS ONE Editorial policies and criteria by including the following statement: "This does not alter our adherence to PLOS ONE policies on sharing data and materials.” as detailed online in our guide for authors  http://journals.plos.org/plosone/s/competing-interests.  If this statement is not true and your adherence to PLOS policies on sharing data and materials is altered, please explain how.Please include the updated Competing Interests Statement and Funding Statement in your cover letter. We will change the online submission form on your behalf.

[I have read the journal's policy and the authors of this manuscript have the following competing interests: AH received a research grant that included matching financial contributions from industry partners for this study JSB received financial compensation from industry partners for his contributions to experimental design and analysis.].   

We note that one or more of the authors are employed by a commercial company: Isabella Point Forestry Ltd.

a.) Please provide an amended Funding Statement declaring this commercial affiliation, as well as a statement regarding the Role of Funders in your study. If the funding organization did not play a role in the study design, data collection and analysis, decision to publish, or preparation of the manuscript and only provided financial support in the form of authors' salaries and/or research materials, please review your statements relating to the author contributions, and ensure you have specifically and accurately indicated the role(s) that these authors had in your study. You can update author roles in the Author Contributions section of the online submission form.

b.) Please also provide an updated Competing Interests Statement declaring this commercial affiliation along with any other relevant declarations relating to employment, consultancy, patents, products in development, or marketed products, etc.  

6. We note that Figure 1, 2 and 3 in your submission contain map/satellite images which may be copyrighted. All PLOS content is published under the Creative Commons Attribution License (CC BY 4.0), which means that the manuscript, images, and Supporting Information files will be freely available online, and any third party is permitted to access, download, copy, distribute, and use these materials in any way, even commercially, with proper attribution. For these reasons, we cannot publish previously copyrighted maps or satellite images created using proprietary data, such as Google software (Google Maps, Street View, and Earth). For more information, see our copyright guidelines: http://journals.plos.org/plosone/s/licenses-and-copyright.

a.)You may seek permission from the original copyright holder of Figure(s) [#] to publish the content specifically under the CC BY 4.0 license. 

b.) If you are unable to obtain permission from the original copyright holder to publish these figures under the CC BY 4.0 license or if the copyright holder’s requirements are incompatible with the CC BY 4.0 license, please either i) remove the figure or ii) supply a replacement figure that complies with the CC BY 4.0 license. Please check copyright information on all replacement figures and update the figure caption with source information. If applicable, please specify in the figure caption text when a figure is similar but not identical to the original image and is therefore for illustrative purposes only.

Reviewers' comments:

Reviewer's Responses to Questions

**Comments to the Author**

1. Is the manuscript technically sound, and do the data support the conclusions?

Reviewer #1: Yes

Reviewer #2: Yes

2. Has the statistical analysis been performed appropriately and rigorously? 

Reviewer #1: Yes

Reviewer #2: I Don't Know

3. Have the authors made all data underlying the findings in their manuscript fully available?

Reviewer #1: Yes

Reviewer #2: Yes

4. Is the manuscript presented in an intelligible fashion and written in standard English?

Reviewer #1: Yes

Reviewer #2: Yes

5. Review Comments to the Author

Reviewer #1: To Authors,

The study by Ding et al., (PONE-19-31818) evaluated ten progeny trails of P. tremultoides in western Alberta, Canada. In doing so, the authors collected survival, height, DHB, leaf abscission data from a subset of test sites planted into southern and northern breeding zones. Overall, the study used roughly 30000 tress to arrive at the conclusions – i.e., growing season is more important for survival and growth than avoiding spring & fall frost.

Line29: Insert “Canada” after Alberta. This helps the readers.

Line31: provide a relevant reference to support aspen short rotation forestry.

Line 35-36: rewrite the sentence – “Selection and breeding…….”

Line 42: change (9) to [9] and fix it throughout the text.

Line 105: replace “gallon” to standard reporting units.

Line 106: provide geographic coordinates of Drayton Valley (latitude, longitude, elevation).

Line 109: provide geographic coordinates of Fairview (latitude, longitude, elevation).

Line 142: replace day of year with “day-of-year” throughout the text.

Line 239: Under Results section….please expand Table 1 to cover survival, bud break and leaf abscission.

Line 262: Expand the results from Table 3.

Line 265: please reconfirm the correlation values to match with Table 4.

Q1. Can authors also address the role of “latitude of origin” on plant height, leaf abscission and breeding value? This would permit to test if photoperiodic adaptation has a role or not.

Line 281: Be specific with fall phenology.

L283-291: I would like to authors to consider moving Fig 2 and Fig 3 description to “Results’ section rather bringing it up under “Discussion” section.

Line 301: Provide relevant reference/s to support your argument.

Line 312: Be specific about leaf abscission and leaf senescence. These are two independent phenological events.

Line 325: Fix typo – trumula to tremula.

Line 337-338: Please confirm the age for growth selection for different sites.

Table 1: Provide units for Lat, Lon, Ele. Also, confirm if its leaf coloration or leaf abscission. Make the changes in Table heading as well.

Table 2: Provide units for height and DBH.

Reviewer #2: This is a well-circumscribed and generally well-written manuscript based on an extensive dataset. The results are interesting but not unexpected. The data do support the main conclusion, that current aspen populations are adapted to colder climates and now lag behind climate warming; but this too must be expected given its lifespan. I do think it’s worthy of publication if the venue is appropriate.

A general problem I have with the discussion is encapsulated by the comment on line 299 that seems to equate growth cessation with leaf abscission, or at least suggests that they are both in the fall. Height growth cessation generally occurs in summer, several weeks before leaf color change even begins. In Populus it takes about 4 weeks for buds to fully form after height growth cessation, then leaf abscission occurs after that. To be sure, there is generally a correlation (at least at the population/provenance level) between date of height growth cessation and date of leaf senescence, but there is greater spread in the former than in the latter. In black cottonwood and balsam poplar from western Canada, height growth cessation generally occurs in mid-July, with buds fully formed by mid-August, and leaf senescence happening in September. Aspen may be different, but probably not hugely so.

The last paragraph seems anti-climactic and disconnected from the title (i.e., the “implications for breeding in a warmer world). Could there be a better ending? What is your take home message?

Other minor points/corrections:

L9 – insert “dates of” before “budbreak”

L15 – Change to “These results suggest” or “This result suggests”

L22 – delete comma after bracket

L25-L26 – I suggest breaking this sentence up into two sentences (after “reproduction”).

L42, and elsewhere – Why do you cite the names of the first two authors on multi-authored papers; i.e., why “St. Clair, Mock (9)” rather than “St. Clair et al. (9)”? Is this correct PLOS One format?

L43 – insert “and” after clones

L45 – change “Heterosis … were” to “Heterosis … was”

L65 – delete hyphens in “within-populations” and “within-families”

L71 – “increase” should be “increases”

L90 – insert “the” before “two breeding regions”

L93 – I suggest deleting “The first two” to start the sentence with “Trials…”

L98 – delete “The” and start sentence with “Trials…”

L106 – most readers will not know that Drayton Valley is in Alberta. Insert AB

L116 – Change “number of half-sib, full-sib families and clones” to “numbers of half-sib and full-sib families, and clones”

L151 – Language needs adjusting here.

L178 – insert “the” after “into”

L194 – language needs adjusting

L211 – change “following” to “follows”?

L283 – I don’t find this surprising. There are generally much larger differences between genotypes/populations for fall/summer phenological traits than for spring phenological traits (in terms of # of days), and the weather is often warmer. A greater effect on the length of the growing season (or photosynthesizing season) is expected, and therefore a greater impact on height and/or survival.

L294 – Relative to positive correlations between productivity and senescence and/or height growth cessation, for Populus trichocarpa see reference 18. For aspen per se (Populus tremula), you might also find interesting a paper by Soolanayakanally et al. (2015. Frontiers in Plant Science 6:528).

[Another thought: I suppose if you have date of budbreak and LAB, could you then take the difference between them as the length of the active photosynthetic period and also then check the correlation with growth? It might be a bit better.]

L325 – “trumula” should be “tremula”

L336 – Was there actual evidence of browse then? More clarity around this point would be useful.

Tables – Throughout the manuscript you refer to the trials by number, not “trial code”. I suggest adding a column for trial number to Table 1 and then using those numbers instead of codes for Tables 2-4.

6. PLOS authors have the option to publish the peer review history of their article (what does this mean?). If published, this will include your full peer review and any attached files.

Reviewer #1: Yes: Raju Soolanayakanahally

Reviewer #2: No

---

## [Author Response · Author response to Decision Letter 0]

30 Jan 2020

Academic Editor’s Comments to the Author:

Ref: PONE-D-19-31818

Dear Dr. DING,

Thank you for submitting your manuscript to PLOS ONE. After careful consideration, we feel that it has merit but does not fully meet PLOS ONE’s publication criteria as it currently stands. Therefore, we invite you to submit a revised version of the manuscript that addresses the points raised during the review process.

Besides meeting the complementary and constructive comments raised by two reviewers, I suggest a bit of effort is put into clarity of tables and figures. I would appreciate it if both origin and planting sites were detailed especially in table 1. It also puzzles me that there is an experimental set up with replicates, but number of replicates (clones) not indicated ... could that number potentially be 1629/6 - or would it be more correct to write 1620 in that column? Fig 2 presents two phenology traits and one growth trait for the northern garden? whereas figure 3 only presents growth traits for the southern garden. Would it make sense to present only phenology responses in figure 2 and then the two height responses (for the southern and northern origin, I believe) could perhaps be collected in the same figure 3?

We would appreciate receiving your revised manuscript by Jan 30 2020 11:59PM. We look forward to receiving your revised manuscript.

Kind regards,

Benedicte Riber Albrectsen

Academic Editor

PLOS ONE

Thank you for your assessment. We made the following changes to address your comments above:

Regarding the use of number of “replicates” in Table 1: what we really mean here is number of “complete blocks”. So, in the “Experimental design” column of this table, 6×10×9×3 means we have 6 complete blocks, 10 incomplete alpha blocks within each complete block, 9 treatments (which can be clone or family) within each incomplete alpha block, and 3 trees in a row plot representing the treatment within an alpha block. So, since we have 3 trees in a non-randomized row plot within an alpha block these are not true “replicates” in a statistical sense. Rather, we have 6 truly independent and randomized experimental units or replicates (i.e. one per complete block). This is a standard design in quantitative genetics, but our description was admittedly a bit confusing. We therefore expanded the footnote 2 to Table 1 as follows:

“2) The experimental design is described as the number of: complete blocks × incomplete alpha blocks within complete blocks × treatments within alpha blocks × trees per treatment in a row plot. The maximum number of treatments (clones or families) in the experiment is determined by the number of alpha bocks × treatments within each alpha block. However, the actual number of tested clones or tested families may be smaller, with filler trees or additional treatment replications filling the gaps.” (Table 1, Page 27)

Yes, that’s a good idea to re-arrange the figures by trait and not by region. That fits better with the narrative as well. Thank you for this suggestion. We changed the titles and captions for Figures 2 and 3 accordingly:

“Fig. 2. Breeding values (BV) of parents on southern breeding region test sites for phenology. The size of the circle indicates the reliability of the estimate (reliability: 0.1 = small circles, 0.7 = large circles).” (Lines 534-536)

“Fig. 3. Height breeding values (BV) of parents tested northern breeding region test sites (left panel) and the southern breeding region sites (right panel). The size of the circle indicates the reliability of the estimate (reliability: 0.1 = small circles, 0.7 = large circles). The color spectrum of red to green indicating low to high BVs, with values ranging from -55 to 45 cm.” (Lines 538-541)

Reviewer #1 Comments

The study by Ding et al., (PONE-19-31818) evaluated ten progeny trails of P. tremultoides in western Alberta, Canada. In doing so, the authors collected survival, height, DHB, leaf abscission data from a subset of test sites planted into southern and northern breeding zones. Overall, the study used roughly 30000 tress to arrive at the conclusions – i.e., growing season is more important for survival and growth than avoiding spring & fall frost.

Line 29: Insert “Canada” after Alberta. This helps the readers.

Changed as suggested. (Line 30). 

Line 31: provide a relevant reference to support aspen short rotation forestry.

We added “Aspen and its hybrids have been utilized in short rotation forestry [6-8].” (Line 29-30)

Line 35-36: rewrite the sentence – “Selection and breeding…….”

Changed to 

“Successful tree selection and breeding programs depend on sufficiently high heritability for traits of commercial interest.” (Lines 35-36)

Line 42: change (9) to [9] and fix it throughout the text.

Corrected throughout the manuscript.

Line 105: replace “gallon” to standard reporting units.

Changed to “1-gallon pots (3.79 liters).” (Line 104-105)

Line 106: provide geographic coordinates of Drayton Valley (latitude, longitude, elevation).

Added: “in Drayton Valley, Alberta (53°13’N, 114°58’W, 869 m).” (Line 105-106)

Line 109: provide geographic coordinates of Fairview (latitude, longitude, elevation).

Added: “Fairview, Alberta (56°04′ N, 118°24′ W, 670 m).” (Line 108-109)

Line 142: replace day of year with “day-of-year” throughout the text.

Changed throughout the manuscript

Line 239: Under Results section….please expand Table 1 to cover survival, bud break and leaf abscission.

Sorry, we are not sure what that comment means. Table 1 already provides survival, bud break and leaf abscission summary statistics, and is referenced in the method. We don’t refer to those traits here because this result section has the subtitle “Genetic parameters of growth traits”.

We also reference this table now under the section “Genetic parameters for adaptive traits”, in case this is what the reviewer meant:

“The phenology traits budbreak and leaf senescence were measured at three trials (Table 1), had moderate broad- and narrow-sense heritabilities (Table 3).” (Line 260) 

Line 262: Expand the results from Table 3.

We added: “Narrow-sense heritabilities for budbreak ranged from 0.4 to 0.5, while heritabilities for leaf senescence were slightly lower with values between 0.3 and 0.4. Broad-sense heritabilities were only marginally higher than narrow-sense heratibilities, and could not be estimated for the seedling trial 2 (Table 3).” (Line 261-264) 

Line 265: please reconfirm the correlation values to match with Table 4.

Corrected: “Moderate to strong genetic correlations were found between growth and phenology (r=-0.3 to -0.2 between height and bud break, and 0.6 to 0.8 between height and leaf senescence)” (Lines 266-269)

Q1. Can authors also address the role of “latitude of origin” on plant height, leaf abscission and breeding value? This would permit to test if photoperiodic adaptation has a role or not.

Our lab did investigate this across larger geographic scales based on range-wide provenance trials (e.g. see Fig. 2 in http://tinyurl.com/t6ena6m). This is a different trial design addressing different questions of population differentiation. Such large-scale geographic patterns would not be detectable in the trials analyzed here, which focus on within-population genetic variation and genetic trait correlations. 

So, we are asking different questions here: The focus of this paper is on genetic correlations among traits to assess potential trade-offs between growth and adaptive traits, and testing the hypothesis that selection for growth may have a correlated response that leads to utilizing a longer growing season, and thereby increases the risk of exposure to late spring frosts or early fall frosts. (i.e., as stated at the end of the introduction)

Line 281: Be specific with fall phenology.

This section of the discussion has been rewritten

L283-291: I would like to authors to consider moving Fig 2 and Fig 3 description to “Results’ section rather bringing it up under “Discussion” section.

OK, that would work as well. We added a final section to the results and also re-arranged the figures as suggested by the editor:

“Trait variation across the landscape

The correlations between fall phenology and height and survival were generally stronger than those between spring phenology and height and survival, indicating that the adaptive value of fall phenology may be surprisingly high. Neither fall phenology nor budbreak showed strong spatial patterns of breeding values across both breeding regions (Fig. 2). However, correlations between height and leaf abscission are visible in these maps: comparing Figs. 2 and 3, high breeding values for height (green dots) are often associated with late leaf senescence (pink and purple), with a Pearson’s correlation coefficient of 0.65 (p<0.0001).” (Lines 274-282)

Line 301: Provide relevant reference/s to support your argument.

Also in response to reviewer #2, we added the following references:

McKown AD, Guy RD, Klapste J, Geraldes A, Friedmann M, Cronk QC, et al. Geographical and environmental gradients shape phenotypic trait variation and genetic structure in Populus trichocarpa. New Phytol. 2014;201(4):1263-76. 

Soolanayakanahally RY, Guy RD, Silim SN, Song M. Timing of photoperiodic competency causes phenological mismatch in balsam poplar (Populus balsamifera L.). Plant Cell Environ. 2013;36(1):116-27. 

Soolanayakanahally R, Guy R, Street N, Robinson K, Silim S, Albrectsen B, et al. Comparative physiology of allopatric Populus species: geographic clines in photosynthesis, height growth, and carbon isotope discrimination in common gardens. Frontiers in Plant Science. 2015;6(528). doi: 10.3389/fpls.2015.00528.

Olson MS, Levsen N, Soolanayakanahally RY, Guy RD, Schroeder WR, Keller SR, et al. The adaptive potential of Populus balsamifera L. to phenology requirements in a warmer global climate. Mol Ecol. 2013;22(5):1214-30. 

D’Orangeville L, Houle D, Duchesne L, Phillips RP, Bergeron Y, Kneeshaw D. Beneficial effects of climate warming on boreal tree growth may be transitory. Nature Communications. 2018;9(1):3213. doi: 10.1038/s41467-018-05705-4.

And the revised section reads:

“In studies with Populus species [34-36] and other species [37], high heritabilities of fall phenology traits, and positive genetic correlations between productivity and fall phenology have previously been observed. Genotypes with a delayed senescence in fall may nevertheless be exposed to increased frost risks. However, climate warming trends that have materialized over the last several decades in Alberta may have decreased the risks of early fall frosts [38]. This might potentially explain our positive association of an extended growing season in fall with high survival. Similar to expectations of Olson et al. [39], genotypes that utilize a longer growing season may be favored by climate warming at northern latitudes. Inadvertent selection, where individuals with better height and diameter growth are chosen, and the genetically correlated leaf senescence is extended in fall, may therefore be an unplanned but effective climate change adaptation strategy. That said, other climatic factors are likely to change, and while the length of time suitable for growth is likely to continue to increase, water limitations may lead to an overall reduction of boreal forest productivity [40,41].” (Lines 294-306)

Line 312: Be specific about leaf abscission and leaf senescence. These are two independent phenological events.

We changed “leaf abscission” to “leaf senescence” throughout the manuscript. Leaf senescence is the most appropriate term for what we measured. We are following the scoring scheme for “autumn senescence” of aspen leaves by Fracheboud et al. (2009), which includes both leaf coloration and leaf abscission criteria.

Line 325: Fix typo – trumula to tremula.

Fixed. (Line 370)

Line 337-338: Please confirm the age for growth selection for different sites.

Changed to “For growth selection at multiple sites, results of >8 growing seasons are adequate for mediocre sites and the productive site.” (Lines 342-343)

Table 1: Provide units for Lat, Lon, Ele. Also, confirm if its leaf coloration or leaf abscission. Make the changes in Table heading as well.

Units for latitude and longitude are given next to the values, and we edited the table heading to specify “Elevation (m)”. We changed “leaf coloration” to “leaf senescence” as explained in reply to the reviewer’s comment on Line 312 above.

Table 2: Provide units for height and DBH.

No, these are heritabilities that do not have units.

Reviewer #2 Comments: 

This is a well-circumscribed and generally well-written manuscript based on an extensive dataset. The results are interesting but not unexpected. The data do support the main conclusion, that current aspen populations are adapted to colder climates and now lag behind climate warming; but this too must be expected given its lifespan. I do think it’s worthy of publication if the venue is appropriate.

A general problem I have with the discussion is encapsulated by the comment on line 299 that seems to equate growth cessation with leaf abscission, or at least suggests that they are both in the fall. Height growth cessation generally occurs in summer, several weeks before leaf color change even begins. In Populus it takes about 4 weeks for buds to fully form after height growth cessation, then leaf abscission occurs after that. To be sure, there is generally a correlation (at least at the population/provenance level) between date of height growth cessation and date of leaf senescence, but there is greater spread in the former than in the latter. In black cottonwood and balsam poplar from western Canada, height growth cessation generally occurs in mid-July, with buds fully formed by mid-August, and leaf senescence happening in September. Aspen may be different, but probably not hugely so.

OK, we clarified that the trait we measure is leaf senescence, and we replaced all instances of “growth cessation” when referring to our own data. We are following the scoring scheme for “autumn senescence” of aspen leaves by Fracheboud et al. (2009), which includes both leaf coloration and leaf abscission criteria. Our metric therefore represents shut-down of photosynthetic activity, and while that timing does not influence the growth of the current season of a plant that has already set but, it will influence overall growth.

The last paragraph seems anti-climactic and disconnected from the title (i.e., the “implications for breeding in a warmer world). Could there be a better ending? What is your take home message?

OK, we followed this paragraph up with a better ending that contains the main take home message, also noted in the abstract.

 “Genetic correlations among fall phenology and spring phenology traits with growth traits were nevertheless already high in this study, and could further increase as the influence of microsite variation of the planting site decreases with the age of the trees. Survival was also not compromised, but was positively associated with early budbreak or late leaf senescence, indicating that the growing season length was more important for survival and growth than avoiding early fall or late spring frosts. We found that a substantial portion of the tested genotypes are adapted to a shorter growing season than they have experienced during the testing period. Selecting genotypes for reforestation that utilize a longer growing season may be unproblematic under continued climate warming in northern latitudes, and may lead to overall increases in forest productivity as long as water availability under increased evapotranspiration does not become a dominant limiting factor. ” (Lines 348-358)

Other minor points/corrections:

L9 – insert “dates of” before “budbreak”

Changed as suggested. (Line 9-10)

L15 – Change to “These results suggest” or “This result suggests”

Fixed. (Line 15).

L22 – delete comma after bracket

Fixed. (Line 22).

L25-L26 – I suggest breaking this sentence up into two sentences (after “reproduction”).

We changed it to “Aspen can regenerate both via sexual and asexual reproduction [2, 3]. Root suckering often produces large single-species stands after fire disturbances in boreal regions [3,4].” (Lines 25-26)

L42, and elsewhere – Why do you cite the names of the first two authors on multi-authored papers; i.e., why “St. Clair, Mock (9)” rather than “St. Clair et al. (9)”? Is this correct PLOS One format?

In text citations have been corrected to “et al.” where applicable, or the sentence structure changed so that spelling out the authors name became unnecessary.

L43 – insert “and” after clones

Added. (Line 41)

L45 – change “Heterosis … were” to “Heterosis … was”

Fixed. (Line 45)

L65 – delete hyphens in “within-populations” and “within-families”

Changed with search and replace throughout the manuscript

L71 – “increase” should be “increases”

Fixed. (Line 70)

L90 – insert “the” before “two breeding regions”

Added. (Line 89)

L93 – I suggest deleting “The first two” to start the sentence with “Trials…”

Changed as suggested. (Line 92)

L98 – delete “The” and start sentence with “Trials…”

Changed as suggested. (Line 97)

L106 – most readers will not know that Drayton Valley is in Alberta. Insert AB

Added. (Line 105)

L116 – Change “number of half-sib, full-sib families and clones” to “numbers of half-sib and full-sib families, and clones”

Changed. (Line 116)

L151 – Language needs adjusting here.

Changed as “For the seedling trials (1 and 2) we employed the following mixed linear model.” (Lines 150-151)

L178 – insert “the” after “into”

Added. (Line 177-178)

L194 – language needs adjusting

Rewritten as “We estimated the additive genetic correlation in seedling trials, genetic correlation in clonal trials (rG), phenotypic correlation (rP) based on individual trees observations.” (Lines 193-194) 

L211 – change “following” to “follows”?

Changed to “… was calculated as follows:” (Line 210)

L283 – I don’t find this surprising. There are generally much larger differences between genotypes/populations for fall/summer phenological traits than for spring phenological traits (in terms of # of days), and the weather is often warmer. A greater effect on the length of the growing season (or photosynthesizing season) is expected, and therefore a greater impact on height and/or survival.

We found the value surprisingly high relative to spring phenology, because frost damage on new leaves should have even greater impact on subsequent growth. In any case, we deleted the word “surprisingly” (Line 278)

L294 – Relative to positive correlations between productivity and senescence and/or height growth cessation, for Populus trichocarpa see reference 18. For aspen per se (Populus tremula), you might also find interesting a paper by Soolanayakanally et al. (2015. Frontiers in Plant Science 6:528).

Also in response to reviewer #1, we added the following references:

McKown AD, Guy RD, Klapste J, Geraldes A, Friedmann M, Cronk QC, et al. Geographical and environmental gradients shape phenotypic trait variation and genetic structure in Populus trichocarpa. New Phytol. 2014;201(4):1263-76. 

Soolanayakanahally RY, Guy RD, Silim SN, Song M. Timing of photoperiodic competency causes phenological mismatch in balsam poplar (Populus balsamifera L.). Plant Cell Environ. 2013;36(1):116-27. 

Soolanayakanahally R, Guy R, Street N, Robinson K, Silim S, Albrectsen B, et al. Comparative physiology of allopatric Populus species: geographic clines in photosynthesis, height growth, and carbon isotope discrimination in common gardens. Frontiers in Plant Science. 2015;6(528). doi: 10.3389/fpls.2015.00528.

Olson MS, Levsen N, Soolanayakanahally RY, Guy RD, Schroeder WR, Keller SR, et al. The adaptive potential of Populus balsamifera L. to phenology requirements in a warmer global climate. Mol Ecol. 2013;22(5):1214-30. 

D’Orangeville L, Houle D, Duchesne L, Phillips RP, Bergeron Y, Kneeshaw D. Beneficial effects of climate warming on boreal tree growth may be transitory. Nature Communications. 2018;9(1):3213. doi: 10.1038/s41467-018-05705-4.

And the revised section reads:

“In studies with Populus species [34-36] and other species [37], high heritabilities of fall phenology traits, and positive genetic correlations between productivity and fall phenology have previously been observed. Genotypes with a delayed senescence in fall may nevertheless be exposed to increased frost risks. However, climate warming trends that have materialized over the last several decades in Alberta may have decreased the risks of early fall frosts [38]. This might potentially explain our positive association of an extended growing season in fall with high survival. Similar to expectations of Olson et al. [39], genotypes that utilize a longer growing season may be favored by climate warming at northern latitudes. Inadvertent selection, where individuals with better height and diameter growth are chosen, and the genetically correlated leaf senescence is extended in fall, may therefore be an unplanned but effective climate change adaptation strategy. That said, other climatic factors are likely to change, and while the length of time suitable for growth is likely to continue to increase, water limitations may lead to an overall reduction of boreal forest productivity [40,41].” (Lines 294-306)

[Another thought: I suppose if you have date of budbreak and LAB, could you then take the difference between them as the length of the active photosynthetic period and also then check the correlation with growth? It might be a bit better.]

Yes, we tried that but the genetic correlation of height with length of the utilized growing season is actually slightly below LAB, although well within the margin of error. We don’t like this composite trait because it is harder to interpret than the individual traits. 

L325 – “trumula” should be “tremula”

Fixed. (Lines 330).

L336 – Was there actual evidence of browse then? More clarity around this point would be useful.

We changed to “and only Trial 7 could not be fenced and showed evidence of browsing, while Trial 8 was located at the more elevated site in the study area.” (Lines 341-342).

Tables – Throughout the manuscript you refer to the trials by number, not “trial code”. I suggest adding a column for trial number to Table 1 and then using those numbers instead of codes for Tables 2-4.

OK, changed as suggested.

---

## [Editor Report · Decision Letter 1]

3 Feb 2020

Genetic parameters of growth and adaptive traits in aspen (Populus tremuloides): implications for tree breeding in a warming world

PONE-D-19-31818R1

Dear Dr. DING,

We are pleased to inform you that your manuscript has been judged scientifically suitable for publication and will be formally accepted for publication once it complies with all outstanding technical requirements.

With kind regards,

Benedicte Riber Albrectsen

Academic Editor

PLOS ONE
---

## [Editor Report · Acceptance letter]

13 Feb 2020

PONE-D-19-31818R1 

Genetic parameters of growth and adaptive traits in aspen (<I>Populus tremuloides</I>): implications for tree breeding in a warming world 

Dear Dr. Ding:

I am pleased to inform you that your manuscript has been deemed suitable for publication in PLOS ONE. Congratulations! Your manuscript is now with our production department. 

With kind regards,

on behalf of

Dr. Benedicte Riber Albrectsen 

Academic Editor

PLOS ONE